# Mouse dead end1 acts with *Nanos2* and *Nanos3* to regulate testicular teratoma incidence

**Atsuki Imai[1], Yoshihiko Hagiwara[1], Yuki Niimi[1¤], Toshinobu Tokumoto[2], Yumiko Saga[3], Atsushi Suzuki** [1,4]*

**1** Division of Materials Science and Chemical Engineering, Graduate School of Engineering, Yokohama National University, Yokohama, Kanagawa, Japan, **2** Biological Science Course, Graduate School of Science, National University Corporation Shizuoka University, Suruga, Shizuoka, Japan, **3** Division of Mammalian Development, National Institute of Genetics, Mishima, Shizuoka, Japan, **4** Division of Materials Science and Chemical Engineering, Faculty of Engineering, Yokohama National University, Yokohama, Kanagawa, Japan

¤ Current address: Center for Exploratory Research, Research & Development Group, Hitachi, Ltd., Kobe, Hyogo, Japan
* suzuki-atsushi-gz@ynu.ac.jp

**Data Availability Statement:** All relevant data are within the manuscript and its Supporting Information files.

**Funding:** This work was supported by the Japan Society for the Promotion of Science (KAKENHI)

## Abstract

Spontaneous testicular teratomas (STTs) derived from primordial germ cells (PGCs) in the mouse embryonic testes predominantly develop in the 129 family inbred strain. *Ter* (spontaneous mutation) is a single nucleotide polymorphism that generates a premature stop codon of Dead end1 (*Dnd1*) and increases the incidence of STTs in the 129 genetic background. We previously found that DND1 interacts with NANOS2 or NANOS3 and that these complexes play a vital role in male embryonic germ cells and adult spermatogonia. However, the following are unclear: (a) whether DND1 works with NANOS2 or NANOS3 to regulate teratoma incidence, and (b) whether *Ter* simply causes *Dnd1* loss or produces a short mutant DND1 protein. In the current study, we newly established a conventional *Dnd1*-knockout mouse line and found that these mice showed phenotypes similar to those of *Ter* mutant mice in spermatogenesis, oogenesis, and teratoma incidence, with a slight difference in spermiogenesis. In addition, we found that the amount of DND1 in *Dnd1*$^{+/Ter}$ embryos decreased to half of that in wild-type embryos, while the expression of the short mutant DND1 was not detected. We also found that double mutants for *Dnd1* and *Nanos2* or *Nanos3* showed synergistic increase in the incidence of STTs. These data support the idea that *Ter* causes *Dnd1* loss, leading to an increase in STT incidence, and that DND1 acts with NANOS2 and NANOS3 to regulate the development of teratoma from PGCs in the 129 genetic background. Thus, our results clarify the role of *Dnd1* in the development of STTs and provide a novel insight into its pathogenic mechanism.

## Introduction

Testicular teratomas are tumors that originate from germ cells. A diverse array of cell and tissue types are found differentiating in these tumors: erythrocyte, adipocyte, cartilage, muscle,

grants 16H01252 and 17H05046 to AS. The funder had no role in study design, data collection and analysis, decision to publish, or preparation of the manuscript.

**Competing interests:** The authors have declared that no competing interests exist.

hair, and glandular tissue, as well as a cluster of stem-like cells from which tumors can be propagated. In mice, spontaneous testicular teratomas (STTs) rarely develop. They predominantly occur in the 129 family of inbred mouse strains; the frequency is 1–7%, depending on the subline [1–3]. In these cases, some primordial germ cells (PGCs) transform to highly proliferative and pluripotent tumor stem cells (embryonal carcinoma [EC] cells) in the embryonic testes at around embryonic day (E) 15.5. Soon after birth, EC cells randomly differentiate into embryonic and adult cells that constitute the tumor in the testes.

In 1973, a spontaneous mutation called *Ter* was isolated, which increased teratoma incidence to 17% in heterozygotes and 94% in homozygous mutants but did not induce ovarian teratomas in the 129/Sv genetic background [4, 5]. In homozygous for *Ter* mutant embryos, the number of PGCs drastically decreases during migration and gonad colonization, partly owing to *Bax*-mediated apoptosis [5–7]. Some of the remaining PGCs in the embryonic testes are thought to give rise to teratomas in the 129/Sv strain. However, in most genetic backgrounds, such as C57BL/6J, LTXBJ, and C3H/HeJ, PGCs disappear until birth and never transform to EC cells, resulting in complete male sterility [8].

In 2005, *Ter* was mapped to Dead end1 (*Dnd1*) [9], a mouse homologue of the zebrafish Dead end gene essential for PGC development [10]. Dead end encodes a vertebrate-specific RNA-binding protein possessing two RNA-recognition motifs (RRMs), among which the second RRM does not conserve three aromatic amino acids playing a key role in nucleic acid–binding activity [11, 12]. In the *Ter* mutation, a single cytosine in the third exon of *Dnd1* is changed to thymine, which generates a premature stop codon (S1A and S1C Fig), presumably resulting in a null mutation of *Dnd1* by nonsense-mediated mRNA decay [13]. Therefore, the defects observed in *Ter* mutant mice have been thought to be attributable to loss of *Dnd1* expression. We have previously shown that DND1 functions with NANOS2 or NANOS3 in both male embryonic germ cells and adult spermatogonia [14, 15], which raises the question of whether DND1 also acts as a partner of NANOS family proteins to regulate the incidence of testicular teratoma. In addition, it was recently reported that a targeted deletion of *Dnd1* did not affect teratoma incidence in the 129S1/SvImJ genetic background; rather, it induced embryonic lethality before E 3.5 [16] (S1B Fig). Thus, these phenotypic differences raised the possibility that the *Ter* mutation generates a short mutant DND1 protein responsible for the phenotype in mutant mice.

In the current study, we first aimed to elucidate the effect of *Dnd1* loss and then to clarify the difference between *Dnd1* loss and *Ter* mutation by comparing our results with previous findings for *Ter* mutant mice. For this purpose, we newly established a conventional knockout mouse line of *Dnd1* and subsequently backcrossed the mutant mice into three different mouse strains: C57BL/6J, MCH(ICR), or 129$^{+Ter}$/Sv (hereafter referred to as BL6, MCH, and 129, respectively). We then examined embryonic lethality, spermatogenesis, oogenesis, and incidence of testicular teratoma for the *Dnd1* mutant mice in these three mouse strains, especially focusing on the 129 strain, and found phenotypes similar to those of *Ter* mutant mice. Additionally, we examined genetic interactions between *Dnd1* and *Nanos2* or *Nanos3* in the regulation of STT incidence and showed that double mutants for *Dnd1* and *Nanos2* or *Nanos3* showed increased incidence of STTs in the 129 genetic background.

## Materials and methods

### Ethics statement

The protocol was approved by the Animal Experiment Committee at Yokohama National University (Project Number: 2019–07; approved on 20 May).

## Mice

We had previously established a *Dnd1*_flox mouse line by using TT2 ES cells and maintained it via intercrosses generating *Dnd1$^{flox/flox}$* mice. To establish the *Dnd1* conventional knockout mouse line, we crossed these *Dnd1$^{flox/flox}$* mice with *Rosa26$^{+/CreERT2}$* mice and obtained *Dnd1$^{+/flox}$; Rosa26$^{+/CreERT2}$* female mice. After administering 75 mg per kg body weight of tamoxifen, we crossed these female mice with wild-type male mice and obtained *Dnd1$^{+/\Delta}$* offspring. The *Dnd1$^{+/\Delta}$* mice were backcrossed into the BL6, 129, or MCH strains for at least eight generations. All three mouse strains were purchased from CLEA Japan (CLEA Japan, Inc., Japan). The 129$^{+Ter}$/SvJcl strain had been established from an STT-high permissive 129/*ter*Sv strain [4] by Dr. Noguchi [5] and provided to and maintained at CLEA Japan, whereas the *Dnd1$^{+/Ter}$* mice had been bred in the animal facility at Shizuoka University by Drs. Tokumoto and Noguchi. Genotyping of *Dnd1*-flox, Δ, and *Ter* alleles was performed as described previously [9, 14]. The *Nanos2$^{+/LacZ}$* and *Nanos3$^{+/Cre}$* mouse lines were established as previously described [17] and backcrossed into the 129$^{+Ter}$/SvJcl strain for at least eight generations.

## Histological methods

For histological analysis, the testes and ovaries were fixed with Bouin's solution and embedded in paraffin. After sectioning (6 μm), the samples were stained with hematoxylin and eosin.

## Tumor surveys

Four-week-old male mice were surveyed for testicular teratomas. Teratoma incidence was calculated as the percentage of male mice with at least one testicular teratoma. Histological analysis (hematoxylin and eosin staining) was used to confirm any teratomas that were ambiguous at autopsy.

## Sperm count

Mature spermatozoa were isolated from the caudal epididymis, as described previously [18]. Briefly, both epididymides from each mouse were minced and incubated in 1 mL warm phosphate-buffered saline (PBS) for 30 min, following which the sperm suspension was fixed in 10% neutral-buffered formalin. The sperm were counted using a hemocytometer. Data have been shown in terms of mean ± standard error of mean (SEM), and the values were statistically analyzed using a Student's *t*-test.

## In vitro fertilization (IVF)

Three-week-old female B6C3F1 mice were purchased from CLEA Japan and superovulated using intraperitoneal injections of 100 μL of CARD HyperOva (Kyudo Co., Ltd., Japan) followed by intraperitoneal injection of 100 μL of 50 units/mL human chorionic gonadotropin (hCG; Aska Pharmaceutical Co., Ltd., Japan) 48 h later. Eggs were recovered 16 to 17 h after hCG injection and placed in a 200 μL CARD MEDIUM (Kyudo Co., Ltd.) drop covered with liquid paraffin (Nacalai Tesque, Inc., Japan) and then used for the experiments 0.5–1 h after preparation. The spermatozoa were collected from the cauda epididymis of more than 12-week-old wild-type or *Dnd1$^{+/\Delta}$* male mice of the 129 strain, suspended in 100 μL CARD FERTIUP (Kyudo Co., Ltd.), and incubated for 1 h for capacitation. Capacitated sperm (3 μL) were added to the drop containing eggs. The eggs were washed in an 80 μL mHTF (Kyudo Co., Ltd.) drop three times after 3 h of coincubation and were transferred to 100 μL KSOM (Kyudo Co., Ltd.) 6 h after insemination. The blastocysts were counted 4 days later.

## Comparison of the PGC number in male gonads at E11.5

Embryonic stage was determined as E11.5 on the basis of the number of somites, while the sex of the embryos was determined by polymerase chain reaction (PCR) for the *Sry* gene using the following primer pairs: Sry-F: 5′-ggttgcaatcataattcttcc-3′ and Sry-R: 5′-cactcctctgtgacactttag-3′.

For section immunostaining, embryonic gonads were fixed with 4% paraformaldehyde overnight at 4˚C and then embedded in paraffin. Whole gonads were sectioned (6 μm) and autoclaved with Antigen Unmasking Solution (Vector Laboratories, Inc., USA). After the samples were subjected to blocking with 5% skim milk in PBS, they were incubated overnight at 4˚C with primary antibodies against deleted in azoospermia-like (DAZL; 1:1,800) [14] and NANOG (1:5,000; IHC-00205, Bethyl Laboratories, Inc., USA) in Can Get Signal immunostain (NKB-501; Toyobo Co., Ltd., Japan). After the samples were washed, they were incubated with Alexa 488- or Alexa 594-conjugated IgG antibodies at 25˚C. The sections were enclosed in Gel/Mount (Biomeda Corp., USA) and observed using fluorescence microscopy (Axio Imager M2, Carl Zeiss, Germany). The number of germ cells in each genital ridge section was assessed by Image J (version 1.50i).

## Western blotting analyses

The 3×Flag expression vectors for *Dnd1* and *Dnd1^{Ter}* were constructed using pcDNA$^{TM}$3.1(+) (Thermo Fisher Scientific, USA). HeLa cells were then transfected with these constructs as previously described [19]. After 48 hours, cellular proteins were extracted with 1×sample buffer (100 mM Tris, pH8.3, 2% SDS, 200 mM DTT, 10% glycerol, 1 mM EDTA, 0.05% bromophenol blue), and then resolved on a 12% sodium dodecyl sulfate (SDS)–polyacrylamide gel electrophoresis (PAGE) gel and electroblotted onto nitrocellulose membrane (BioTrace NT, Pall Corporation, USA). The membranes were incubated with primary antibodies; anti-FLAG antibody (1:8,000, F3165, Sigma-Aldrich), or anti-DND1 antibodies (1:1,000) generated from rabbit and guinea pig [20]. These were followed by goat anti-mouse IgG conjugated with alkaline phosphatase (AP; 1:2,000, 69266, Novagen) for anti-FLAG antibody, swine anti-rabbit IgG conjugated with AP (1:2,000, D0306, DAKO) for rabbit anti-DND1 antibody, or goat anti-guinea pig IgG conjugated with AP (1:2,000, sc-2930, Santa Cruz Biotechnology) for guinea pig anti-DND1 antibody. The detection of immunoreactivity was performed using a BCIP/NBT Phosphatase substrate kit (50-81-00, SeraCare Life Sciences, Inc., USA) according to the manufacturer's instructions.

For western blotting analyses of proteins from E15.5 male gonads, embryonic gonads were excised from E15.5 male embryos from pregnant *Dnd1^{+/Ter}* female mice crossed with wild-type male mice and then sonicated with an ultrasonic disruptor (Handy Sonic UR-20P; Tomy Seiko Co., Ltd., Japan.) in 1×sample buffer. Extracts were resolved on 12.5% SDS-PAGE gels and electroblotted onto nitrocellulose membranes (BioTrace NT, Pall Corporation). The membranes were incubated with primary antibodies: anti-DND1 antibody generated by guinea pig #2 (1:1,000) [14] or anti-DAZL antibody generated from rabbits (1:1,000) [20], followed by goat anti-guinea pig IgG conjugated with horseradish peroxidase (HRP; 1:10,000; ab7139, Abcam plc, UK) or goat anti-rabbit IgG conjugated with HRP (1:10,000, sc-2054, Santa Cruz Biotechnology, USA)). Immunoreactivities were visualized as chemiluminescence by using Western BLoT Chemiluminescence HRP Substrate (Takara Bio Inc., Japan) and a lumino-image analyzer (ImageQuant LAS-4000mini, GE Healthcare, England) and then quantitated according to the manufacturer's instructions.

All antibodies used in western blotting analysis were diluted by Can Get Signal Immunoreaction Enhancer Solution (NKB-101; Toyobo Co., Ltd.).

## Flow cytometry

Testis cell suspensions were generated by sequential digestion of dissected and minced semi-niferous tubules with 1 mg/mL collagenase (Wako, Japan), followed by 0.25% trypsin (27250018; Thermo Fisher Scientific) containing 1 mM EDTA. Cells were passed through a 40-μm strainer to remove clumps. Harvested cells were resuspended in HBSS containing 5% fetal bovine serum (FBS) and 0.1% bovine serum albumin (BSA). Subsequently, a FoxP3 Tran-scription Factor Staining Buffer Kit (A25866A; Life Technologies, USA) was used with an anti-body against promyelocytic leukemia zinc-finger (PLZF; 1:200; H-300; Santa Cruz Biotechnology) according to the manufacturer's instructions. DNA was labelled with 4′,6-dia-midino-2-phenylindole (DAPI) to gate the 2N and 4N fractions. Stained cells were analyzed with a cell sorter (MoFlo Astrios; Beckman Coulter, USA), and data analysis was performed using KALUZA (version 1.2, Beckman Coulter).

## Statistical analysis

The statistical significance of differences in the ratios of testis weight per body weight, ratios of defective tubules, sperm count, and relative DND1 expression were assessed using a two-tailed *t*-test, whereas the data for genotype distribution in the intercrosses of 129 $Dnd1^{+/\Delta}$ heterozy-gotes and tumor surveys were analyzed with $\chi^2$ analysis and Fisher's exact test, respectively. A *P*-value of <0.05 was considered to represent statistical significance. Statistical analysis was performed using Microsoft Excel for Mac (version 16.33) or R (version 3.6.0).

## Results

### Homozygous for *Dnd1*-Δ mutant mice were born at Mendelian ratios and developed testicular teratoma in the 129 genetic background

To clarify the effect of *Dnd1* loss, we aimed to establish a new conventional *Dnd1*-knockout mouse line. We had previously generated a *Dnd1*-flox mouse line to conditionally remove DND1 by Cre recombinase [14]. In the current study, a conventional knockout (*Dnd1*-Δ) mouse line was generated from the *Dnd1*-flox mice by Cre-mediated excision of exons 2 and 3, which included the two RRMs of DND1 (S1D and S1E Fig), and was subsequently backcrossed into three different mouse strains: BL6, MCH, or 129. Intercrosses of $Dnd1^{+/\Delta}$ heterozygotes within each strain produced the $Dnd1^{\Delta/\Delta}$ homozygotes in accordance with a Mendelian expec-tation of 1:2:1 in all three mouse strains (Table 1), showing no evidence for embryonic lethality of the homozygous mutants even in the 129 genetic background.

We then examined whether the *Dnd1*-Δ allele affects the occurrence of testicular teratoma. For this purpose, testes from 4-week-old $Dnd1^{\Delta/\Delta}$ mice, $Dnd1^{+/\Delta}$ mice, and their $Dnd1^{+/+}$ wild-type control littermates of all three strains were harvested and surveyed for teratomas (Table 2 and S2 Fig). In the 129 strain, the *Dnd1*-Δ allele significantly increased teratoma

**Table 1. Genotype distribution of offspring from $Dnd1^{+/\Delta}$ intercrosses in BL6, MCH, and 129 strains.**

|  | Genotype of *Dnd1* | | | Total no. examined | $x^2$ | P |  |
|---|---|---|---|---|---|---|---|
|  | +/+ | +/Δ | Δ/Δ |  |  |  |  |
| BL6 | 46 | 82 | 43 | 171 | 0.39 | 0.82 | ns |
| MCH | 93 | 171 | 97 | 361 | 1.09 | 0.58 | ns |
| 129 | 102 | 178 | 95 | 375 | 1.22 | 0.54 | ns |

The number of $Dnd1^{\Delta/\Delta}$ progeny derived from $Dnd1^{+/\Delta}$ crosses were compared among BL6, MCH, and 129 strains. Adult offspring of both sexes were genotyped. *ns: not significnat.

**Table 2. Genotype of *Dnd1* and the incidence of male mice affected with testicular teratomas in the BL6, MCH, and 129 strains.**

| BL6 | % of affected males | | | |
|---|---|---|---|---|
| Genotype | total* | unilateral (L) | unilateral (R) | bilateral |
| +/+ | 0% (0/32) | - | - | - |
| +/Δ | 0% (0/36) | - | - | - |
| Δ/Δ | 0% (0/20) | - | - | - |
| MCH | % of affected males | | | |
| Genotype | total* | unilateral (L) | unilateral (R) | bilateral |
| +/+ | 0% (0/47) | - | - | - |
| +/Δ | 0% (0/85) | - | - | - |
| Δ/Δ | 10.2% (6/59) | 5.1% (3/59) | 1.7% (1/59) | 3.4% (2/59) |
| 129 | % of affected males | | | |
| Genotype | total* | unilateral (L) | unilateral (R) | bilateral |
| +/+ | 4.2% (4/97) | 3.1% (3/97) | 1.0% (1/97) | - |
| +/Δ | 28.8% (42/146) | 14.4% (21/146) | 7.5% (11/146) | 6.8% (10/146) |
| Δ/Δ | 92.6% (50/54) | 9.3% (5/54) | 20.4% (11/54) | 63.0% (34/54) |

*Dnd1*$^{+/Δ}$ female mice were crossed with *Dnd1*$^{+/Δ}$ or *Dnd1*$^{+/+}$ male mice among the BL6, MCH, and 129 strains to test the incidence of male offspring with at least one testicular teratoma.

*Fisher's exact test: MCH-*Dnd1*$^{+/Δ}$ versus MCH-*Dnd1*$^{Δ/Δ}$, $P = 0.00404$; 129-*Dnd1*$^{+/+}$ versus 129-*Dnd1*$^{+/Δ}$, $P = 5.28E\text{-}07$; 129-*Dnd1*$^{+/Δ}$ versus 129-*Dnd1*$^{Δ/Δ}$, $P = 2.2E\text{-}16$

occurrence in male mice from a baseline of 4.2% to 28.8% in *Dnd1*$^{+/Δ}$ heterozygotes and 92.6% in *Dnd1*$^{Δ/Δ}$ homozygotes, which are ratios similar to those for *Ter* mutant mice [4, 5, 16, 21]. In contrast, teratomas developed in the testes of approximately 10% of the MCH *Dnd1*$^{Δ/Δ}$ male mice, indicating that the MCH strain has low sensitivity to testicular teratoma, while teratomas were not observed in the testes of all three genotypes of the BL6 strain. Collectively, the results indicate that the *Dnd1*-Δ allele is inherited in accordance with Mendelian ratios, similar to the findings for the *Dnd1*-*Ter* allele, and that the ratios of testicular teratoma incidence are similar between the Δ and *Ter* alleles.

### *Dnd1* loss caused defects in both spermatogenesis and oogenesis

We also examined the effect of *Dnd1* loss on germ cell development. For this purpose, we first compared testes from 4-week-old *Dnd1*$^{+/+}$ wild-type, *Dnd1*$^{+/Δ}$, and *Dnd1*$^{Δ/Δ}$ male mice that did not have testicular teratomas. We found that even testes of *Dnd1*$^{+/Δ}$ mice appeared to be smaller than those of wild-type male mice, in addition to a clear decrease in testis size noted in *Dnd1*$^{Δ/Δ}$ male mice in all three strains (Fig 1A–1C). The ratios of testis weight per body weight of *Dnd1*$^{+/Δ}$ male mice were significantly lower than those of wild-type male mice in all three strains, while the ratios of *Dnd1*$^{Δ/Δ}$ male mice showed a drastic decrease compared with those of wild-type and *Dnd1*$^{+/Δ}$ male mice (Fig 1D). Histological analyses showed some seminiferous tubules with impaired spermatogenesis in *Dnd1*$^{+/Δ}$ testes, while no germ cells were observed in *Dnd1*$^{Δ/Δ}$ testes (Fig 1E–1M). We counted the tubules with impaired spermatogenesis in *Dnd1*$^{+/Δ}$ testes and found significant increases in the ratio of defective tubules as compared with those of wild-type testes in all three strains, especially in the 129 strain (Fig 1N–1P). These data indicate that *Dnd1* loss results in disappearance of germ cells in adult testes and that spermatogenesis is impaired in a part of the seminiferous tubules, even in *Dnd1*$^{+/Δ}$ heterozygous male mice.

We then compared the ovaries of 4-week-old female mice of each genotype among the three strains (Fig 2A–2C) and found that the *Dnd1*$^{+/Δ}$ ovaries were as large as the wild-type ovaries in

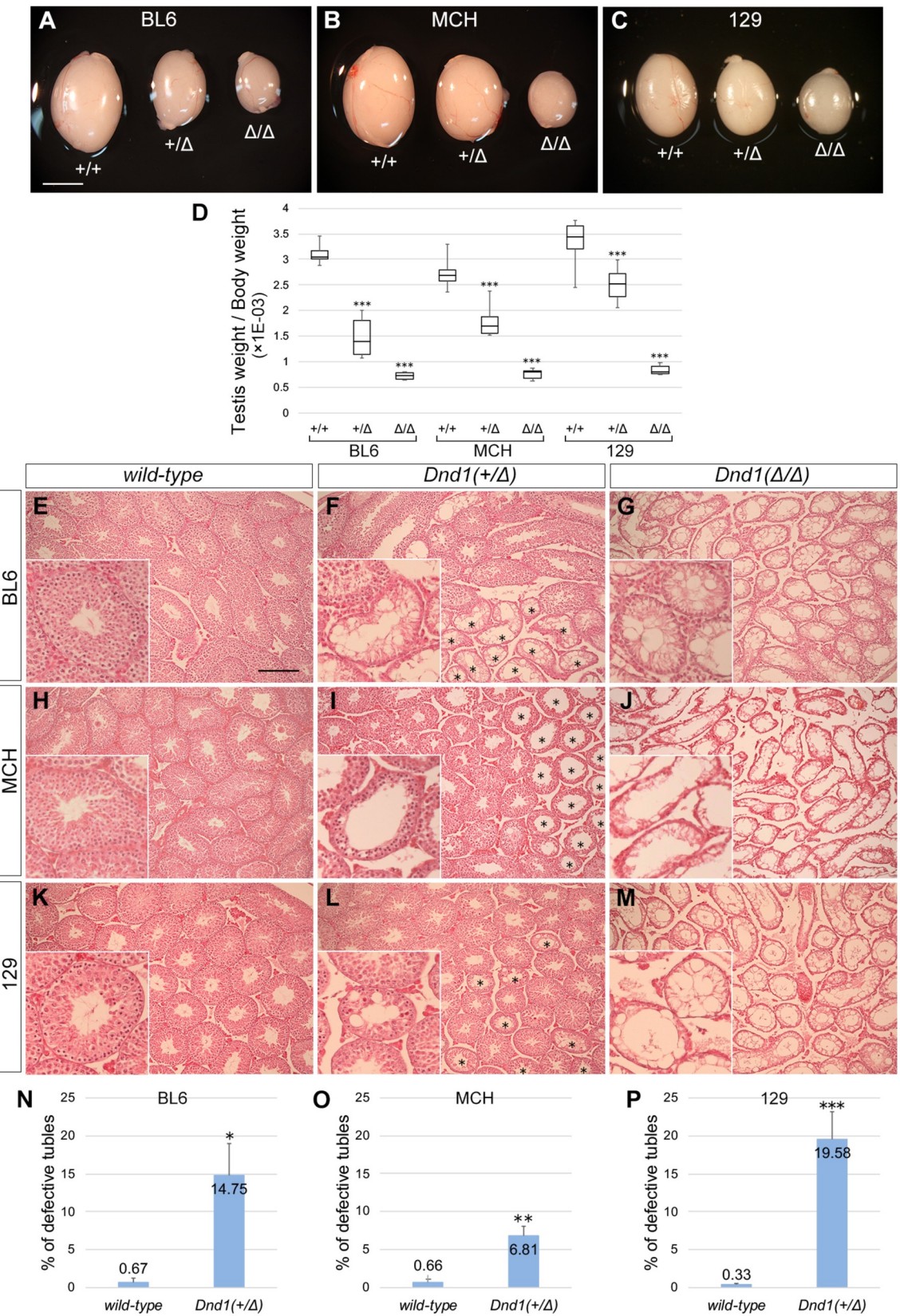

**Fig 1. Comparison of testicular phenotype of *Dnd1-Δ* mutant male mice among the BL6, MCH, and 129 strains.** (A–C) Comparison of the testis size of 4-week-old littermates of wild-type, *Dnd1*$^{+/\Delta}$, and *Dnd1*$^{\Delta/\Delta}$ mice of the BL6 (A), MCH (B), and 129 (C) strains. Scale bar: 5 mm in A for A–C. (D) Box plots (median [horizontal line], 25th and 75th percentiles [box], and maximum and minimum [error bars]) indicate the testis weight per body weight of male mice not affected with teratomas. ***$P < 0.001$ (Student's *t*-test). (E–M) Testis sections of 4-week-old littermates of wild-type (E, H, K), *Dnd1*$^{+/\Delta}$ (F, I, L), and *Dnd1*$^{\Delta/\Delta}$ (G, J, M) mice of the BL6 (E, F, G), MCH (H, I, J), and 129 (K, L, M) strains were stained with hematoxylin and eosin. Asterisks indicate tubules with defective spermatogenesis in *Dnd1*$^{+/\Delta}$ testes (F, I, L). Insets show enlarged views to better visualize tubules. Scale bar: 200 μm in E for E–M. (N–P) Comparison of percentages of seminiferous tubules with defective spermatogenesis among wild-type and *Dnd1*$^{+/\Delta}$ mice of the BL6 (N), MCH (O), and 129 (P) strains; more than 180 tubules from three independent cross-sections of the testes were scored ($n = 3$). *$P < 0.01$, **$P < 0.005$, ***$P < 0.001$ (Student's *t*-test).

all the three strains, in contrast to the findings for testes. Only the *Dnd1*$^{\Delta/\Delta}$ ovaries from the 129 female mice appeared to be slightly larger than those of the other two strains, although the ovaries from *Dnd1*$^{\Delta/\Delta}$ female mice were smaller than those of wild-type and *Dnd1*$^{+/\Delta}$ female mice in all three strains. Histological analyses showed that a considerable number of oocytes were developing in the ovaries from 129 *Dnd1*$^{\Delta/\Delta}$ female mice, whereas the other two strains had few oocytes in *Dnd1*$^{\Delta/\Delta}$ ovaries (Fig 2D–2L). We then crossed these female mice at 6 weeks of age with MCH male mice and counted the number of offspring to examine their fertility until they reached 30 weeks of age (Fig 2M). These analyses showed no significant difference between wild-type and *Dnd1*$^{+/\Delta}$ female mice in each strain; no offspring were born from *Dnd1*$^{\Delta/\Delta}$ female mice in both BL6 and MCH strains. However, the 129 female mice delivered offspring even in the case of *Dnd1*$^{\Delta/\Delta}$ mice, but the number was fewer than those of wild-type and *Dnd1*$^{+/\Delta}$ female mice, indicating that 129 *Dnd1*$^{\Delta/\Delta}$ female mice were subfertile.

Collectively, the data indicate that the spermatogenic and oogenic phenotypes observed in *Dnd1-Δ* mutant mice were very similar to those previously reported for *Ter* mutant mice [5].

## *Dnd1*$^{+/\Delta}$ male mice progressively lost fertility because of sperm count decrease and impaired sperm function in the 129 strain

We next checked whether the *Dnd1-Δ* mutant mice exhibited phenotypes other than those mentioned above. We noted that 129 *Dnd1*$^{+/\Delta}$ male mice produced fewer offspring than BL6 and MCH strains in their intercrosses. For more in-depth analysis of the phenotype, we crossed 10-week-old *Dnd1*$^{+/\Delta}$ male mice of each strain with female mice of the MCH strain and determined the litter size. BL6 or MCH *Dnd1*$^{+/\Delta}$ male mice continuously impregnated the female mice and produced offspring until they were 20–24 weeks old (Fig 3A and 3B). In contrast, female mice that were crossed with 129 *Dnd1*$^{+/\Delta}$ male mice produced offspring until the male mice reached 10–12 weeks of age, but the number of offspring drastically decreased after 12 weeks of age (Fig 3C). Subsequently, they stopped giving birth when the male mice were at 20–24 weeks of age; however, copulatory plugs were continuously found in the female mice. Since wild-type 129 male mice could impregnate the female mice even after they were 12 weeks old (Fig 3D), this phenotype was caused by the heterozygous *Dnd1-Δ* mutant allele only in the 129 genetic background.

To identify the cause of infertility in 129 *Dnd1*$^{+/\Delta}$ male mice, we compared the ratios of testis weight per body weight of 4-week-old wild-type and *Dnd1*$^{+/\Delta}$ male mice with those of 12-week-old male mice in each of the three strains. The ratios of *Dnd1*$^{+/\Delta}$ male mice were lower than those of wild-type male mice at both time points and in all three strains (Fig 3E–3G), consistent with the results shown in Fig 1D. Furthermore, we found that the ratio significantly decreased from 4 to 12 weeks of age in only 129 *Dnd1*$^{+/\Delta}$ male mice, whereas in the other two strains, the ratio increased even in *Dnd1*$^{+/\Delta}$ male mice, suggesting testicular growth retardation due to impaired spermatogenesis. We therefore determined the sperm count of 12-week-old wild-type and *Dnd1*$^{+/\Delta}$ male mice in each strain and found that the count had significantly decreased in *Dnd1*$^{+/\Delta}$ male

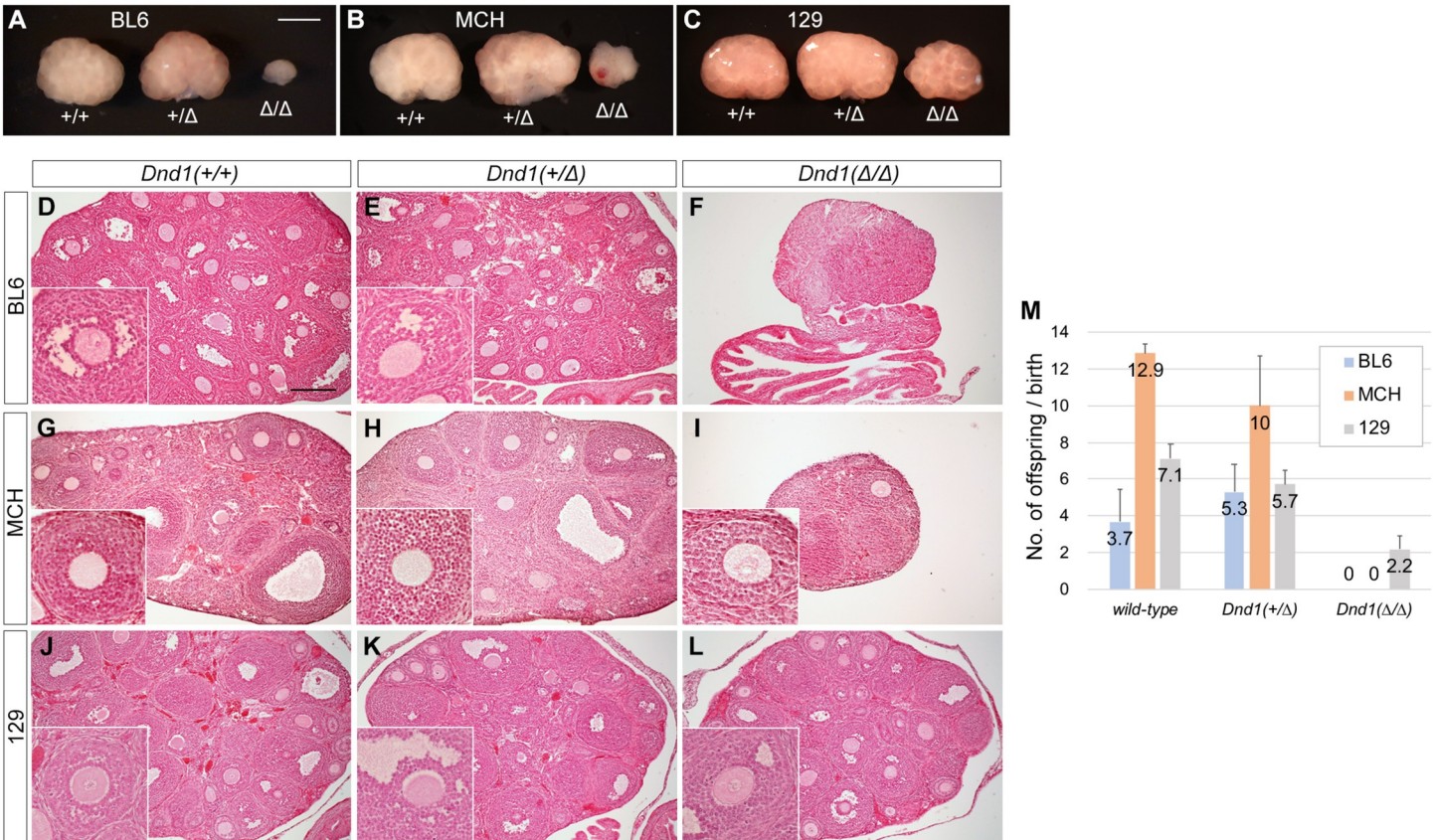

**Fig 2. Comparison of ovarian phenotype of *Dnd1*-Δ mutant female mice among the MCH, BL6, and 129 strains.** (A–C) Comparison of ovary size of 4-week-old littermates of wild-type, *Dnd1*$^{+/Δ}$, and *Dnd1*$^{Δ/Δ}$ mice of the BL6 (A), MCH (B), and 129 (C) strains. Scale bar: 1 mm in A for A–C. (D–L) Ovary sections of 4-week-old littermates of wild-type (D, G, J), *Dnd1*$^{+/Δ}$ (E, H, K), and *Dnd1*$^{Δ/Δ}$ (F, I, L) mice of the BL6 (D, E, F), MCH (G, H, I), and 129 (J, K, L) strains were stained with hematoxylin and eosin. Insets show enlarged views to better visualize oocytes. Note that there are no oocytes in F. Scale bar: 200 μm in D for D–L. (M) Litter size analysis of wild-type, *Dnd1*$^{+/Δ}$, and *Dnd1*$^{Δ/Δ}$ female mice of the BL6, MCH, and 129 strains (*n* = 3).

mice of all the strains. However, the sperm counts of 129 *Dnd1*$^{+/Δ}$ male mice decreased to approximately one-fifth of those of wild-type male mice (Fig 3J), whereas the decrease was only approximately one-half in the other two strains (Fig 3H, 3I), suggesting that such a large reduction in sperm count might lead to fertility loss in 129 *Dnd1*$^{+/Δ}$ male mice. However, since the sperm count reduction does not necessarily cause fertility loss [22], we further examined whether the sperm generated in 129 *Dnd1*$^{+/Δ}$ male mice were functional by performing IVF with sperm from more than 12-week-old 129 wild-type or *Dnd1*$^{+/Δ}$ male mice (Fig 3K). Nearly 90% of oocytes developed to blastocysts when sperm from wild-type male mice were used. However, in the case of sperm from *Dnd1*$^{+/Δ}$ male mice, the developmental rate drastically decreased to 3.9%, indicating that the sperm were not fully functional in 12-week-old 129 *Dnd1*$^{+/Δ}$ male mice. Collectively, the findings indicate that the 129 *Dnd1*$^{+/Δ}$ male mice lost fertility, presumably because of a significant decrease in sperm count and impaired function of sperm after 12 weeks of age.

To determine why a large decrease in sperm count and impaired sperm function occurred only in 129 *Dnd1*$^{+/Δ}$ male mice after 12 weeks of age, we speculated that the impaired spermatogenesis causing these phenotypes might be attributable to some defect in spermatogonia because *Dnd1* is expressed in spermatogonia and is required for the maintenance of these cells [15]. To examine this hypothesis, cells in the testes from 12-week-old wild-type and *Dnd1*$^{+/Δ}$ male mice of all three strains were subjected to flow cytometric analysis performed using

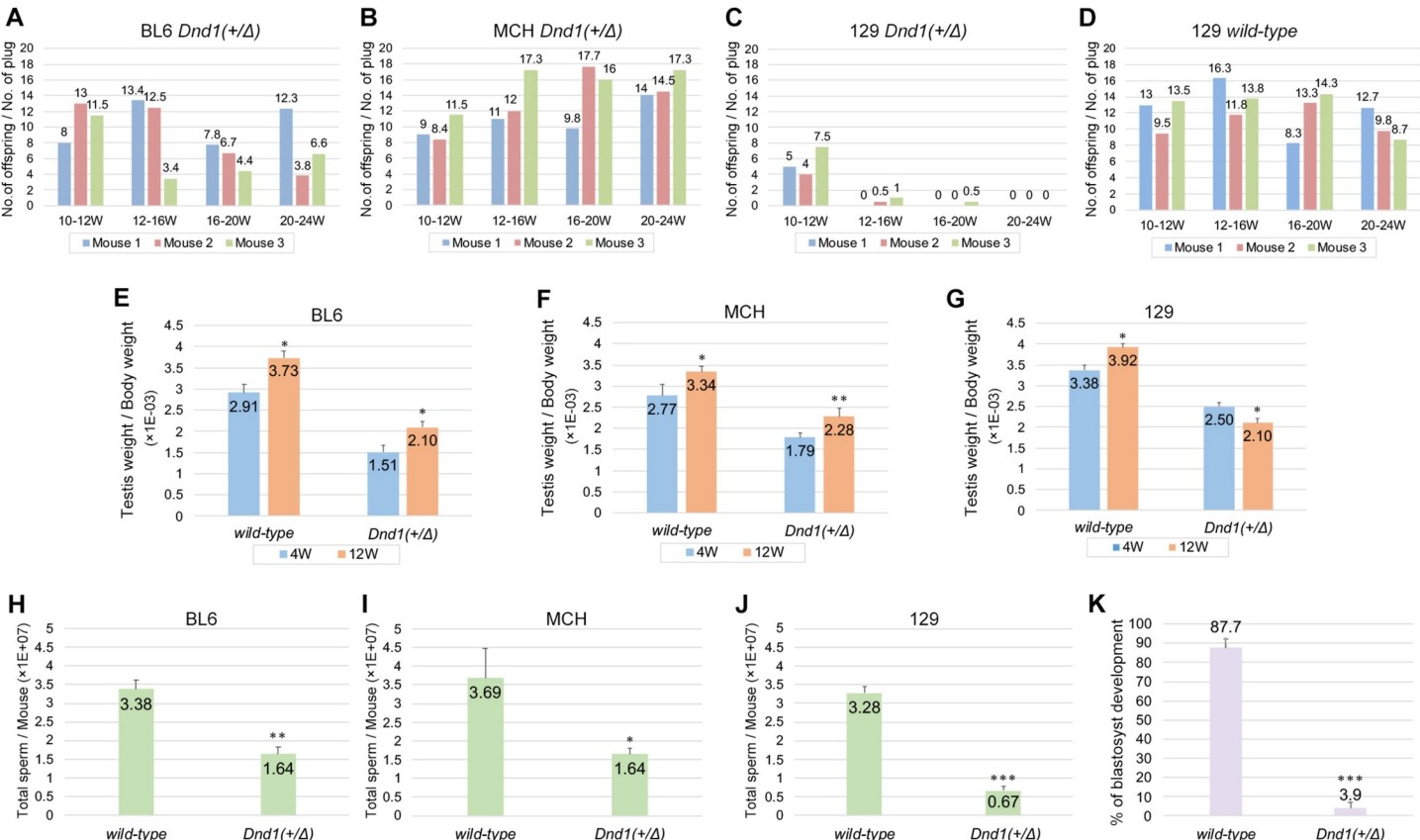

**Fig 3. 129 *Dnd1*<sup>+/Δ</sup> male mice progressively lost fertility because of sperm count decrease and sperm malfunction.** (A–D) Litter size analysis of *Dnd1*<sup>+/Δ</sup> male mice of the BL6, MCH, and 129 strains and wild-type male mice of the 129 strain. A 10-week-old male mouse was crossed with three wild-type female MCH mice until the male mouse reached 24 weeks of age. Three male mice (Mouse 1, Mouse 2, and Mouse 3) were analyzed per strain or genotype. (E–G) Comparison of testis weight per body weight ratios between 4-week-old and 12-week-old wild-type and *Dnd1*<sup>+/Δ</sup> mice from the BL6, MCH, and 129 strains. Error bars represent mean ± SD; three mice were analyzed per genotype and strain. $^{*}P < 0.01$, $^{**}P < 0.005$ (Student's $t$-test). (H–J) Sperm count analysis of 12-week-old wild-type and *Dnd1*<sup>+/Δ</sup> male mice of the BL6, MCH, and 129 strains. Error bars represent mean ± SD; three mice were analyzed per genotype and strain. $^{*}P < 0.01$, $^{**}P < 0.005$, $^{***}P < 0.001$ (Student's $t$-test). (K) IVF analysis using sperm from more than 12-week-old wild-type and *Dnd1*<sup>+/Δ</sup> male mice of the 129 strain. Error bars represent mean ± SD; three mice were analyzed per genotype. $^{***}P < 0.001$ (Student's $t$-test).

antibodies against promyelocytic leukemia zinc-finger (PLZF) because we have previously shown that PLZF is expressed in all populations of DND1-positive spermatogonia [15]. These analyses revealed that the number of PLZF-positive spermatogonia significantly decreased in both 129 and BL6 *Dnd1*<sup>+/Δ</sup> male mice (S3A–S3F Fig). However, although BL6 *Dnd1*<sup>+/Δ</sup> male mice showed a slightly greater decrease in the number of PLZF-positive spermatogonia than 129 *Dnd1*<sup>+/Δ</sup> male mice did (S3C and S3F Fig), the decrease in sperm count in BL6 *Dnd1*<sup>+/Δ</sup> male mice was much lesser than that in 129 *Dnd1*<sup>+/Δ</sup> male mice (3H and J). Furthermore, although the number of spermatogonia was unchanged between wild-type and *Dnd1*<sup>+/Δ</sup> male mice in the MCH strain (S3G–S3I Fig), the sperm count significantly decreased in *Dnd1*<sup>+/Δ</sup> male mice (Fig 3I). These data indicate no correlation between the decrease in spermatogonia and the decrease in sperm count in *Dnd1*<sup>+/Δ</sup> male mice.

## Progressive loss of fertility did not occur in 129 *Dnd1*<sup>+/Ter</sup> male mice

In relation to testicular teratoma incidence, spermatogenesis, and oogenesis, the phenotypes of *Dnd1*-Δ mutant mice were similar to those of *Ter* mutant mice [5]. However, the progressive

loss of fertility observed in 129 *Dnd1*$^{+/\Delta}$ male mice has not been previously reported in 129 *Dnd1*$^{+/Ter}$ male mice. To determine whether the *Ter* mutant allele leads to the same phenotype for fertility as the *Dnd1-\Delta* mutant allele in the 129 strain, we compared the ratio of testis weight per body weight between 12-week-old wild-type and *Dnd1*$^{+/Ter}$ male mice in the 129 strain and found a significant decrease in the ratio to 70% of that of wild-type male mice in the case of *Dnd1*$^{+/Ter}$ male mice (Fig 4A). However, the reduction rate was lesser than that of *Dnd1*$^{+/\Delta}$ male mice since the *Dnd1-\Delta* mutant allele decreased the ratio by almost half (Fig 3G). We therefore compared the sperm count of 12-week-old 129 *Dnd1*$^{+/Ter}$ male mice with those of wild-type male mice and found that the number tended to decrease but was not significantly changed as compared with that of wild-type male mice (Fig 4B). We then crossed 129 *Dnd1*$^{+/Ter}$ male mice with female mice of the MCH strain and determined the litter size, as mentioned in Fig 3, to check whether the *Ter* mutant allele caused progressive loss of fertility, as is the case with 129 *Dnd1*$^{+/\Delta}$ male mice (Fig 3C). These analyses revealed that 129 *Dnd1*$^{+/Ter}$ male mice continuously impregnated the female mice and produced offspring until they were 20–24

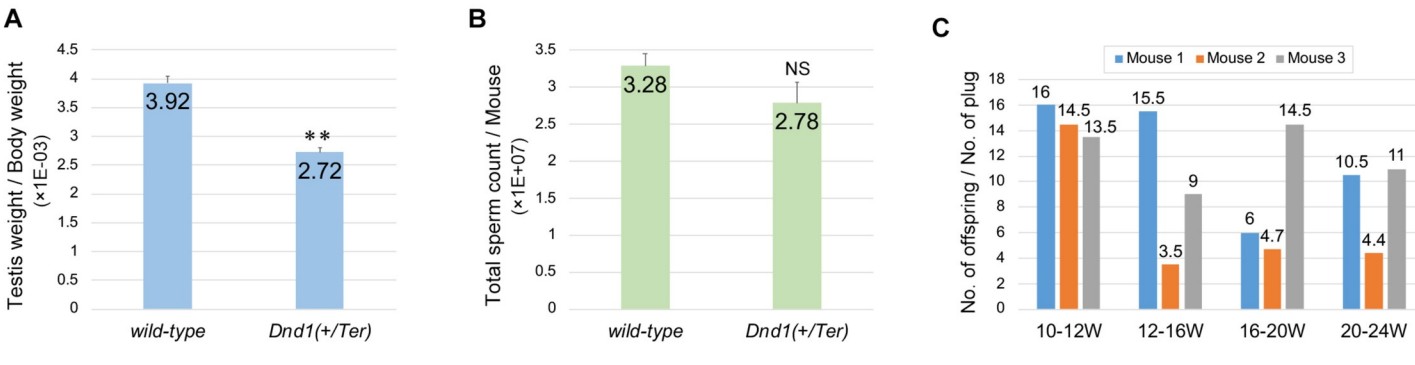

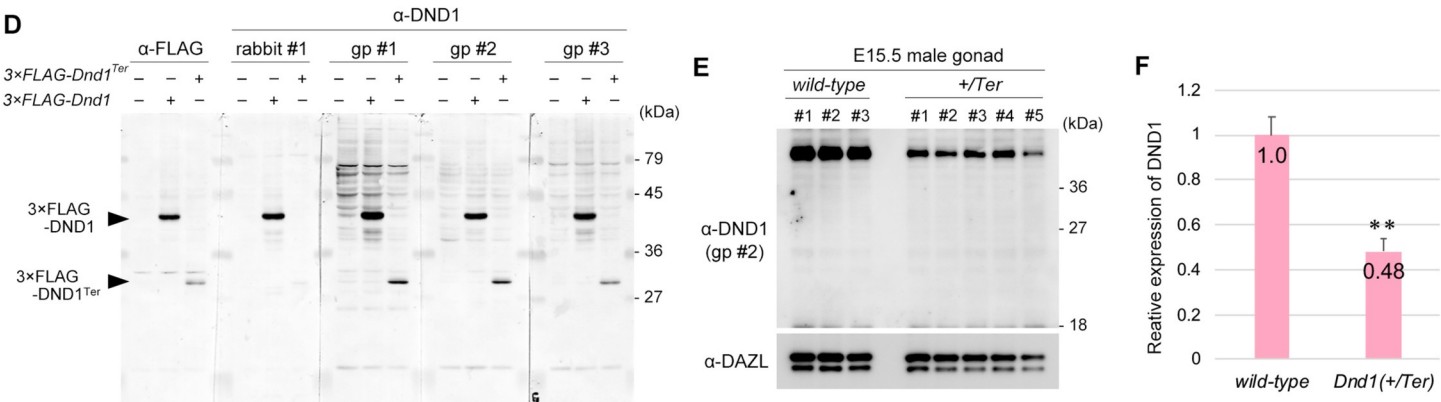

**Fig 4. *Dnd1*$^{+/Ter}$ mutant male mice maintained fertility and did not produce a short mutant DND1 protein.** (A) Testis weight per body weight ratios of 12-week-old wild-type and *Dnd1*$^{+/Ter}$ mice of the 129 strain were measured. Error bars represent mean ± SD; three mice were analyzed per genotype. **$P < 0.005$ (Student's *t*-test). (B) Sperm count analysis of 12-week-old wild-type and *Dnd1*$^{+/Ter}$ male mice of the 129 strain. Error bars represent mean ± SD; three mice were analyzed per genotype. (C) Litter size analysis of *Dnd1*$^{+/Ter}$ male mice of the 129 strain. A 10-week-old *Dnd1*$^{+/Ter}$ male mouse was crossed with three wild-type female MCH mice until the male mouse was 24 weeks of age. Three male mice (Mouse 1, Mouse 2, and Mouse 3) were analyzed. (D) Western blotting analyses of Flag-tagged wild-type DND1 and DND1$^{Ter}$ in HeLa cells transfected with FLAG-tagged *Dnd1*, *Dnd1*$^{Ter}$, or control vector by using anti-DND1 antibodies generated by rabbit #1, guinea pig #1 (gp #1), guinea pig #2 (gp #2), and guinea pig #3 (gp #3). (E) Western blotting analyses of proteins from E15.5 male gonads of wild-type and *Dnd1*$^{+/Ter}$ embryos, performed using anti-DND1 antibodies generated by guinea pig #2 (gp #2). DAZL is a loading control. Note that no specific bands were observed in all five *Dnd1*$^{+/Ter}$ male gonads at approximately 27 kDa. (F) Comparison of the DND1 relative expression level in wild-type and *Dnd1*$^{+/Ter}$ E15.5 male gonads. The signal intensities of DND1 in (E) were normalized by those of DAZL. Error bars represent mean ± SD; **$P < 0.005$ (Student's *t*-test).

weeks old, unlike the 129 *Dnd1*$^{+/\Delta}$ male mice (Fig 4C). These data thus showed a clear pheno-typic difference between *Dnd1*$^{+/\Delta}$ and *Dnd1*$^{+/Ter}$ male mice in the 129 strain, suggesting that either the *Dnd1-Δ* or *Ter* mutant allele caused some gene expression change, other than *Dnd1* loss, and that such changes generated these phenotypic differences.

In this context, we focused on the *Ter* mutant allele because it was previously suggested that *Ter* was not a mutation causing loss of *Dnd1* expression, but instead generated a short mutant DND1 protein, called DND1$^{Ter}$, consisting of *Dnd1* exons 1–2 and a part of exon 3 (S1C Fig) [16, 23]. To examine DND1$^{Ter}$ expression, we first tested whether antibodies against DND1, which we previously generated from a rabbit and three guinea pigs [14], could detect DND1$^{Ter}$. For this purpose, wild-type DND1 or DND1$^{Ter}$ were forcibly expressed in HeLa cells by transfection of 3×Flag-tagged *Dnd1* or *Dnd1*$^{Ter}$ and then subjected to western blot using four different antibodies against DND1. Our analyses showed that the antibodies gener-ated from guinea pigs #1, #2, and #3 could detect Flag-tagged DND1$^{Ter}$ at the same level as the anti-FLAG antibody, while the antibody generated from rabbit #1 barely detected it (Fig 4D). We then analyzed the *in vivo* expression of DND1$^{Ter}$ in the E15.5 male gonads of 129 *Dnd1*$^{+/Ter}$ embryos by using the antibody against DND1 generated from guinea pig #2. However, western blot analyses revealed that DND1$^{Ter}$ was undetectable in all five *Dnd1*$^{+/Ter}$ embryos (Fig 4E), whereas the DND1 levels decreased to approximately half of those in wild-type embryos (Fig 4F). These results support the possibility that *Ter* is a loss of *Dnd1* mutation, which in turn suggests that the *Dnd1-Δ* mutant allele causes some defect leading to sperm count decrease and sperm malfunction independently of *Dnd1* loss.

## *Dnd1* genetically interacted with *Nanos2* and *Nanos3* for suppression of testicular teratomas

We have previously shown that both NANOS2 and NANOS3 interact with DND1 and that these complexes are essential for germ cell development [14, 15], leading us to speculate that both NANOS2 and NANOS3 play a vital role with DND1 even in the regulation of testicular teratoma incidence. To examine this hypothesis, we crossed *Dnd1-Δ* mutant mice with both *Nanos2* or *Nanos3* mutant mice to establish male mice that were double mutants for *Dnd1* and *Nanos2* or *Nanos3* and subsequently analyzed these mice for testicular teratomas in the 129 genetic background.

In the case of crossing with *Nanos2* mutant mice (Figs 5A and S4A), the ratio of affected male mice slightly increased from 2.9% in wild-type male mice to 8.8% in *Nanos2*$^{+/LacZ}$ male mice and 11.8% in *Nanos2*$^{LacZ/LacZ}$ male mice, indicating that *Nanos2* is one of the genes responsible for testicular teratomas, as previously reported [24]. In addition, introducing the heterozygous *Dnd1-Δ* mutant allele into both *Nanos2*$^{+/LacZ}$ and *Nanos2*$^{LacZ/LacZ}$ male mice increased the ratio to 38.8% and 80.0%, respectively, indicating a synergistic effect of combin-ing mutations for *Dnd1* and *Nanos2* since only 28.5% (Table 2) of *Dnd1*$^{+/\Delta}$ male mice were affected. In the case of crossing with *Nanos3* mutant mice (Figs 5B and S4B), the ratio of affected male mice increased from 4.4% in wild-type male mice to 24.2% in *Nanos3*$^{+/Cre}$ male mice, indicating that *Nanos3* is one of the genes responsible for testicular teratomas, as previ-ously reported [25]. In addition, the male mice that were double mutants for *Dnd1*$^{+/\Delta}$; *Nanos3*$^{+/Cre}$ showed a drastic increase in the ratio to 84.3%, indicating a synergistic effect of combining heterozygous mutations for *Dnd1* and *Nanos3*. Based on these data, we conclude that both *Nanos2* and *Nanos3* interact with *Dnd1* in the regulation of testicular teratoma incidence.

Testicular teratomas were not observed in *Nanos3*$^{Cre/Cre}$ male mice, even when these mice had a heterozygous *Dnd1-Δ* mutant allele. To determine why *Nanos3*$^{Cre/Cre}$ male mice were

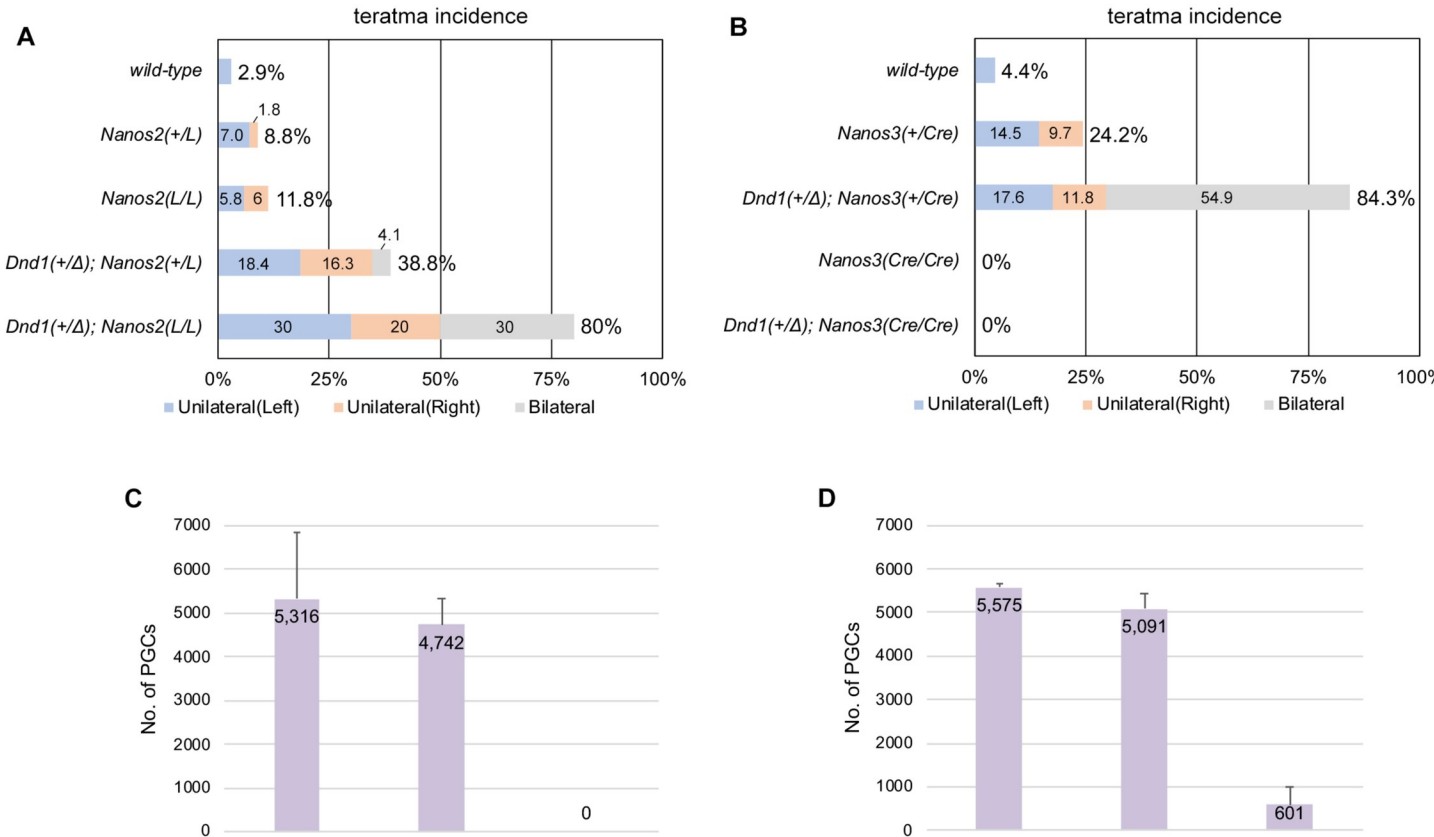

**Fig 5. Both *Nanos2* and *Nanos3* are involved in the regulation of *Dnd1*-mediated teratoma susceptibility.** (A, B) Incidence of testicular teratomas in 129 male mice carrying the *Dnd1*-Δ allele and *Nanos2*-LacZ (A) or *Nanos3*-Cre (B) allele. The numbers of male mice examined were as follows: *wild-type* (N = 35), *Nanos2^{+/LacZ}* (N = 57), *Nanos2^{LacZ/LacZ}* (N = 17), *Dnd1^{+/Δ}*; *Nanos2^{+/LacZ}* (N = 49), and *Dnd1^{+/Δ}*; *Nanos2 ^{LacZ/LacZ}* (N = 10) in (A), and *wild-type* (N = 45), *Nanos3^{+/Cre}* (N = 62), *Dnd1^{+/Δ}*; *Nanos3^{+/Cre}* (N = 51), *Nanos3^{Cre/Cre}* (N = 30), and *Dnd1^{+/Δ}*; *Nanos3^{Cre/Cre}* (N = 20) in (B). Blue, orange, and gray boxes in (A) and (B) indicate the percentages of cases where left, right, and both testes, respectively, were affected. (C, D) Comparison of the PGC number among wild-type, *Nanos3^{+/Cre}*, and *Nanos3^{Cre/Cre}* (C) or wild-type, *Dnd1^{+/Δ}*, and *Dnd1^{Δ/Δ}* (D) male embryos at E11.5.

not affected, we counted the number of PGCs in *Nanos3*-Cre mutant male embryos at E11.5 by section immunostaining of gonads with antibodies against DAZL and NANOG and then compared the numbers with those in *Dnd1*-Δ mutant male embryos. These analyses revealed that PGCs were almost completely absent in *Nanos3^{Cre/Cre}* embryos at E11.5 (Fig 5C), which is before the transformation to EC cells. This might be because the PGCs undergo apoptotic cell death [26] or transdifferentiate into somatic cells as observed in other species [27, 28] during migration stages in the absence of *Nanos3*. In contrast, a small number of PGCs were still observed in the male gonads of *Dnd1^{Δ/Δ}* embryos (Fig 5D), suggesting that some of these survivor cells eventually acquire pluripotency at the late stages of embryogenesis and then develop to form teratomas after birth.

## Discussion

In the current study, we newly generated *Dnd1*-Δ, a conventional knockout allele of *Dnd1* (S1E Fig), and found that the *Dnd1*-Δ mutant mice showed defects similar to those of *Ter* mutant mice in spermatogenesis, oogenesis, and incidence of testicular teratomas, with a slight difference in spermiogenesis. In addition, a short mutant DND1 protein, called DND1^{Ter}, was

not detected in E15.5 male gonads from *Dnd1*$^{+/Ter}$ embryos in our western blotting analysis (Fig 4D and 4E), whereas the amounts of full-length DND1 in *Dnd1*$^{+/Ter}$ embryos decreased to about half of those in wild-type embryos (Fig 4F). On the basis of these data, we suggest that the *Ter* mutation simply causes loss of *Dnd1* expression, as previously reported [9]. However, we cannot rule out the possibility that DND1$^{Ter}$ is highly expressed in germ cells other than at E15.5 or that DND1$^{Ter}$ is translated but is undetectable because its expression level is too low. The expression level of DND1$^{Ter}$ was lesser than that of wild-type DND1 when both proteins were force-expressed in cultured cells (Fig 4D), which was consistent with a previous study suggesting that DND1$^{Ter}$ was unstable [23]. In either case, DND1$^{Ter}$ might be involved in the development of testicular teratomas. Alternatively, given that the defects in *Dnd1-Ter* mutant mice were milder than those in *Dnd1-Δ* mutant mice (Figs 3J and 4B), it is also possible that DND1$^{Ter}$ alleviates the phenotypes of *Dnd1* loss, thereby leading to the phenotypic differences in sperm count between *Dnd1-Δ* and *Ter* mutant mice.

The 129 *Dnd1*$^{+/Δ}$ male mice lost fertility after 12 weeks because of sperm count decrease and impaired sperm function, which was not observed in 129 *Dnd1*$^{+/Ter}$ male mice (Figs 3 and 4C). If *Ter* is a loss of *Dnd1* mutation as suggested, these phenotypic differences between *Dnd1*$^{+/Δ}$ and *Dnd1*$^{+/Ter}$ might be caused by sequence deletion in the *Dnd1-Δ* allele. Generating the *Dnd1-Δ* allele might collaterally remove regulatory sequence(s) located between two loxP sequences controlling the expression of gene(s) around *Dnd1*, which might cause changes in the expression of other gene(s), rather than loss of *Dnd1*. This is because *Dnd1* is located in a genomic region where 16 genes are closely placed within 150 kilobase pairs, thereby resulting in sperm count decrease and impaired sperm function. However, it is unclear which gene expression level is affected by the targeted deletion and which of the cells are affected by these gene expression changes occurring with the *Dnd1-Δ* mutant allele because no study has reported the physiological role of the 15 neighboring genes in spermatogenesis (S5 Fig). Since no correlation was observed between the decrease in spermatogonia and the decrease in sperm count in *Dnd1*$^{+/Δ}$ male mice (S3 Fig), the sperm count decrease and impaired sperm function could be attributed to a defect in cells other than spermatogonia.

In the context of phenotypic difference, Zechel et al. reported that, in the 129S1/SvImJ genetic background, homozygous deletion from *Dnd1* exon 1 to most of exon 3 (S1B Fig) induced embryonic lethality before E3.5 [16]. In addition, they showed that the incidence of testicular teratomas did not increase in the 129S1/SvImJ *Dnd1*$^{+/KO}$ heterozygous mutant male mice although *Dnd1*$^{+/Ter}$ male mice showed increased incidence in the same genetic background. These phenotypes are different from those of both *Dnd1-Δ* and *Ter* mutant mice. As mentioned above, the deletion from *Dnd1* exon 1 to most of exon 3 might collaterally remove regulatory sequence(s) that control the expression of genes placed close to *Dnd1*, leading to phenotypes different from those in *Dnd1-Δ* and *Ter* mutant mice. Alternatively, the phenotypic difference might be caused by the neomycin selector unit because it is widely known that insertion of a pgk-neo cassette or reporter gene into a specific locus might affect the expression of the neighboring genes [29, 30]. It is also possible that a de novo mutation occurred during establishment of the *Dnd1*-KO mouse line and caused changes in the expression of the neighboring genes, resulting in the phenotypic differences. The expression profile and phenotypes of mice with knockout of the genes around *Dnd1* would yield valuable information for further understanding the phenotypic differences among *Dnd1-Δ*, *Ter*, and *Dnd1*-KO mutant mice.

We have also shown synergistic increase in testicular teratoma incidence in the male mice that were double mutants for *Dnd1* and *Nanos2* or *Nanos3* (Fig 5A and 5B). In addition to these genetic interactions, given that both NANOS2 and NANOS3 associate with DND1, it is highly plausible that both DND1-NANOS2 and DND1-NANOS3 complexes regulate teratoma incidence in male embryonic germ cells. We have previously shown that DND1 plays a role in

loading RNAs onto the NANOS2-CNOT deadenylase complex, leading to degradation of the target RNAs [15]. In addition, in the absence of NANOS2, a part of male embryonic germ cells continuously proliferate and cannot undergo cell-cycle arrest [31], which increases the incidence of transformation of germ cells to EC cells in the 129 strain [24]. These results suggest that the DND1-NANOS2 complex suppresses target RNAs that induce cell-cycle arrest via CNOT deadenylase–mediated RNA degradation, thereby suppressing the transformation of germ cells to EC cells. NANOS3 is highly expressed in migrating and proliferative PGCs but its expression gradually disappears after colonization to male gonads, in accordance with the increase in NANOS2 expression [17, 32]. Therefore, the regulatory mechanisms of the DND1-NANOS3 complex in the case of testicular teratoma incidence are still unknown and may differ from those of NANOS2. Comprehensive identification of target RNAs for DND1-NANOS2 and DND1-NANOS3 complexes is required to elucidate the transformation of germ cells to EC cells.

## Supporting information

**S1 Fig. Comparison of *Dnd1* genome structure.** Structures of the *Dnd1* (A-E) wild-type allele (A), KO allele generated by Zechel et al. [16] (B), *Ter* allele [9] (C), floxed allele generated by Suzuki et al. [14] (D), and Δ allele (E).
(PDF)

**S2 Fig. Testicular teratomas in the *Dnd1*-Δ mutant mice of the 129 strain.** Comparison of the testes from 4-week-old $Dnd1^{+/\Delta}$ and $Dnd1^{\Delta/\Delta}$ mice of the 129 strain. Note that testicular teratomas developed only in the right testis in the $Dnd1^{+/\Delta}$ mice, whereas both testes have testicular teratoma in the $Dnd1^{\Delta/\Delta}$ mice. Scale bar: 5 mm.
(PDF)

**S3 Fig. Reduction in spermatogonial number is not related to sperm count decrease and impaired sperm function.** Representative flow cytometric analyses of testis cells from 12-week-old wild-type (A, D, G) and $Dnd1^{+/\Delta}$ (B, E, H) mice of the 129 (A–C), BL6 (D–F), and MCH (G–I) strains for PLZF. Percentages of cells within each PLZF-positive gate (A, B, D, E, F, G) were normalized by the ratio of PLZF-positive cells from wild-type mice and indicated (C, F, I). Error bars represent mean ± SD; three mice were analyzed per genotype and strain. $^{*}P < 0.05$ (Student's *t*-test).
(PDF)

**S4 Fig. Testicular teratomas in mice that were double mutants for *Dnd1* and *Nanos2* or *Nanos3* in the 129 strain.** Comparison of the testes from 4-week-old $Dnd1^{+/\Delta}; Nanos2^{+/LacZ}$ and $Dnd1^{+/\Delta}; Nanos2^{LacZ/LacZ}$ mice (A), or wild-type and $Dnd1^{+/\Delta}; Nanos3^{+/Cre}$ mice of the 129 strain (B). Scale bars: 5 mm in (A) and (B).
(PDF)

**S5 Fig. The 15 genes neighboring *Dnd1* and their knockout mouse lines.** The 15 genes neighboring *Dnd1* are listed in order of their location on chromosome 18. Their knockout mouse (KO) lines and phenotypes are also mentioned in the list. IMPC: The International Mouse Phenotyping Consortium.
(PDF)

**S1 Raw images.**
(JPG)

## Acknowledgments

We thank H. Nishimura for her technical assistance with the tumor survey.

## Author Contributions

**Conceptualization:** Atsushi Suzuki.

**Data curation:** Atsuki Imai, Yoshihiko Hagiwara, Yuki Niimi.

**Funding acquisition:** Atsushi Suzuki.

**Investigation:** Atsuki Imai, Yoshihiko Hagiwara, Yuki Niimi, Atsushi Suzuki.

**Project administration:** Atsushi Suzuki.

**Resources:** Toshinobu Tokumoto, Yumiko Saga.

**Writing – original draft:** Atsushi Suzuki.

**Writing – review & editing:** Atsushi Suzuki.

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
