## [Decision Letter · Decision Letter 0]

26 Feb 2020

PONE-D-20-03327

Dead end1 acts with Nanos2 and Nanos3 to regulate testicular teratoma incidence

PLOS ONE

Dear Dr. Suzuki,

Thank you for submitting your manuscript to PLOS ONE. After careful consideration, we feel that it has merit but does not fully meet PLOS ONE’s publication criteria as it currently stands. Therefore, we invite you to submit a revised version of the manuscript that addresses the points raised during the review process.

The three referees are delighted to see a valid and well written study. I agree with this statement and consider the findings highly relevant to be published. The reviewers raise several questions and suggest a few changes which may improve the paper. The authors should comment to these concerns and incorporate the appropriate changes. I consider the revision minor.

We would appreciate receiving your revised manuscript by Apr 11 2020 11:59PM. To enhance the reproducibility of your results, we recommend that if applicable you deposit your laboratory protocols in protocols.io, where a protocol can be assigned its own identifier (DOI) such that it can be cited independently in the future. For instructions see: http://journals.plos.org/plosone/s/submission-guidelines#loc-laboratory-protocols

We look forward to receiving your revised manuscript.

Kind regards,

Stefan Schlatt

Academic Editor

PLOS ONE

Journal Requirements:

Reviewers' comments:

Reviewer's Responses to Questions

**Comments to the Author**

1. Is the manuscript technically sound, and do the data support the conclusions?

Reviewer #1: Yes

Reviewer #2: Yes

Reviewer #3: Yes

2. Has the statistical analysis been performed appropriately and rigorously? 

Reviewer #1: Yes

Reviewer #2: Yes

Reviewer #3: Yes

3. Have the authors made all data underlying the findings in their manuscript fully available?

Reviewer #1: Yes

Reviewer #2: Yes

Reviewer #3: Yes

4. Is the manuscript presented in an intelligible fashion and written in standard English?

Reviewer #1: Yes

Reviewer #2: Yes

Reviewer #3: No

5. Review Comments to the Author

Reviewer #1: General comments:

In this study, Imai and Colleagues followed on their previous work on the mechanisms of action of Dnd1 in germ cells of 129/Sv mice, which frequently develop spontaneous testicular teratomas (STT). In the current study they examined the consequences of Dnd1 loss and looked at interactions between DND1 and NANOS2 and NANOS3.

The authors established a dnd1-KO mouse line and back-crossed it to three different mouse strains characterised by very different frequencies of SST, including a highly permissive 129/terSv and a resistant C57BL/6J strain. Furthermore, they crossed the 120+ter/SvJcl strain with either Nanos2- or Nanos3- manipulated strains. The detailed analysis of the phenotypes comprised incidence of teratomas, fertility, and embryonic and postnatal testis and ovary histology, with germ cell quantification, including analysis of flow-sorted PLZF-positive germ cells. Germ cell recognition / analysis of gene expression was done by IHC (DAZL, NANOG) and quantitative analysis of studied proteins (DND1; DAZL) by western blotting. The authors reported a significant rise in the incidence of SST only in 129/Dnd1- mice, with rates similar to Ter mice. Cross-breeding experiments with Nanos2- or Nanos3- showed that both these genes are involved in the origin of teratomas in mice. In all strains they found decreased adult testis size, significant germ cell loss in dnd1-heterozygotes and a complete germ cell loss in the homozygotes. The sperm quality decreased with age, but not in the 129ter. Oogenesis and fertility were affected in all dnd1- strains, with the mildest phenotype in 129dnd1- strain.

The results concerning the phenotypic differences between 129ter and 129dnd1- as well as the cross-breeding experiments with Nanos2- or Nanos3- are novel and shed light on the mechanisms of STT pathogenesis. Although the conclusion that ter mutation causes dndl1 loss confirms previous studies, this is the most rigorous confirmation of this hypothesis. The work is very comprehensive and technically very well done. The figures are very nice and informative. The findings are discussed carefully, without over-interpreting the data.

I congratulate the authors for their excellent work and think that the paper is ready for publication essentially as is.

Minor specific comments:

1. The studied species (mouse) should be mentioned in the title.

2. The description of figures is included in the text of the Results and there are no separate figure legends. In my opinion this avoids redundant data repeating but the editors can decide whether or not this is OK.

Reviewer #2: Imai et al. introduce a new Dnd1Δ/Δ mouse line with a loss of function phenotype and show that double mutants of Dnd1 with Nanos2 and Nanos3 increase the testicular teratoma incidence. By comparing their knockout phenotype in different genetic backgrounds to the well-described Dnd1+/Ter mice, they show similarities in the phenotypes supporting the hypothesis that the Ter mutation leads to a loss of functional protein. By using different genetic backgrounds, they address the known influence of the genetic background on the DND1 mutations on the tumor incidence.

The authors present a well designed study. They showed appropriate controls and their conclusions are supported by statistical analyses. All methods are described in sufficient detail.

Minor points for improvement prior to publication:

- Figure presentation: The labeling of the graphs is relatively small and may be very hard to read when considering the usual figure size in the formatted manuscript.

- Line 105: What is the genetic background of the DND1flox/flox used to produce the Dnd1Δ/Δ mouse line? The cited publication (Suzuki et al. 2016) does not describe this specific line in detail.

- Line 526: In your discussion, you mention the observed differences to the study of Zechel et al. While I appreciate the given explanations, I was wondering how you ensured that the backcrosses still share the 100% of the genetics and that no other (de novo) mutations occurred explaining the striking differences between Dnd1Δ/Δ and Dnd1KO/KO mice?

- Line 540: You mention genes close to DND1, that may be regulated by deleted regions and potentially explain observed differences. Your point may be strengthened by giving specific examples, for example if neighboring genes play a known role in developmental processes affected or if the genetic organization (TADs, known enhancers) in this region are described in the literature and do confirm your hypothesis.

Reviewer #3: The paper “Dead end1 acts with Nanos2 and Nanos3 to regulate testicular teratoma incidence” by Imai et al present functional analysis of the Dead end protein in mouse.

The authors present the phenotype of the “full knockout” of the gene and compare it with that observed in the Ter mutation where the protein is truncated.

The authors suggest only the development of the germline is affect in the mutant, which is different from the conclusion of a previous paper where Dead end was considered to be essential for embryonic development. The authors provide evidence that from the point of view of teratoma development the complete null and the Ter allele are similar. The authors also present the phenotype Dead end loss of function has on ovary development, which is a poorly studied topic. Last, the authors present a series of experiments should genetic interactions between Dead end and Nanos proteins, thereby contributing to the understanding of the molecular mechanisms of Dead end function.

Overall, the paper provides new data that is of importance for researchers in the field of reproductive biology and the work should definitely be published in the journal after addressing the following points.

1. The paper would benefit a lot from English editing. In many cases the message is understood only after reading certain sentences several times.

2. “In the BL6 strain, teratomas were not observed in the testes of all three genotypes, whereas teratomas developed in the testes of approximately 10% of the MCH Dnd1Δ/Δ male mice, indicating that the MCH strain has low sensitivity to testicular teratoma.” – reading this sentence one should would define the MCH strain as having HIGH sensitivity, since the comparison is to the BL6 strain that does not form teratomas. The authors should modify / correct this sentence.

3. In Figure 1A- C the authors present examples for the testes phenotypes. The least severe difference between the wild-type and the mutant is observed in Figure 1C (for the 129 strain). From Figure 1D, the 129 strain shows the strongest effect. The authors should present in 1A-C examples that fit the average phenotype presented in Figure 1D.

4. In Figure 1E-M, show magnified boxes in all cases not only in panels F, I and L. This way the phenotypes can be better appreciated.

5. “Only the Dnd1Δ/Δ ovaries from the 129 female mice appeared to be slightly larger than those of the other two strains…”. If this data exists, provide quantitative information rather than “slightly”.

6. In Figures 2D-L, provide magnification boxes and marked specific cell types / structures such that the information is more accessible for a broader readership.

7. In Figure 3A, B, mark the mice as Nr 1 , Nr 2 and Nr 3 rather than No 1, No 2 and No 3. As it is, it looks like a gene name and not a mouse number. The best would be to use Mouse 1 , Mouse 2 and Mouse 3 as there is enough space for that.

8. The authors should try to reduce the text in the figure legends such that the information is not redundant with what is provided in the figure itself. For example in the legend of Figure 3- “The vertical axis represents the average number of offspring per copulatory plug, while the horizontal axis represents the age of the male mice.” The same information is provided in the Figure itself.

9. “…were subjected to flow cytometric analysis performed using antibodies against PLZF, because… “ define PLZF in the text the first time it is mentioned and not only in the methods section.

10. The authors start a section with the sentence “We next checked whether the Dnd1-Δ mutant mice exhibited phenotypes other than those of Ter mutant mice.”. In this section the Ter mutant is not directly referred to. The authors should rephrase it.

11. “These results support the idea that Ter is a loss of Dnd1 mutation, which in turn suggests that the Dnd1-Δ mutant allele causes some defect leading to sperm count decrease and sperm malfunction independently of Dnd1 loss.” This statement is very strong and is not proven by the data provided. The assumption that Ter is a complete loss of function, which questions the specificity of the knockout phenotype relies on Western blots. As very low amounts of the shorter version of Dead end are not necessarily detected, it could in principle be that the milder phenotype of Ter results from low amounts of the truncated version of the protein. In addition, it could be that the level of the Ter protein is higher at stages that differ from those examined in the Western blots. This point should be addressed and the statement the authors put forward should either be deleted, or presented as one of several options.

12. “In addition, introducing the heterozygous Dnd1-Δ mutant allele into both Nanos2+/LacZ and Nanos2LacZ/LacZ male mice increased the ratio to 38.8% and 80.0% (Fig. S4A)..” – the information is not. Provided in Figure S4A. Perhaps the authors mean Figure 5a here.

13. In Figure 5 the lack of germ cells in homozygous nanos3 mutants is determined by expression of daz and nanog. The authors conclude that the cells underwent apoptosis, but do not show it directly. The loss of cells could for example be transfating etc. The sentence should just be rephrased or the apoptosis claim deleted.

6. PLOS authors have the option to publish the peer review history of their article (what does this mean?). If published, this will include your full peer review and any attached files.

Reviewer #1: No

Reviewer #2: No

Reviewer #3: Yes: Erez Raz and Kim Westerich

---

## [Author Response · Author response to Decision Letter 0]

26 Mar 2020

Responses to reviewers’ comments

Reviewer #1: General comments:

In this study, Imai and Colleagues followed on their previous work on the mechanisms of action of Dnd1 in germ cells of 129/Sv mice, which frequently develop spontaneous testicular teratomas (STT). In the current study they examined the consequences of Dnd1 loss and looked at interactions between DND1 and NANOS2 and NANOS3.

The authors established a dnd1-KO mouse line and back-crossed it to three different mouse strains characterised by very different frequencies of SST, including a highly permissive 129/terSv and a resistant C57BL/6J strain. Furthermore, they crossed the 120+ter/SvJcl strain with either Nanos2- or Nanos3- manipulated strains. The detailed analysis of the phenotypes comprised incidence of teratomas, fertility, and embryonic and postnatal testis and ovary histology, with germ cell quantification, including analysis of flow-sorted PLZF-positive germ cells. Germ cell recognition / analysis of gene expression was done by IHC (DAZL, NANOG) and quantitative analysis of studied proteins (DND1; DAZL) by western blotting. The authors reported a significant rise in the incidence of SST only in 129/Dnd1- mice, with rates similar to Ter mice. Cross-breeding experiments with Nanos2- or Nanos3- showed that both these genes are involved in the origin of teratomas in mice. In all strains they found decreased adult testis size, significant germ cell loss in dnd1-heterozygotes and a complete germ cell loss in the homozygotes. The sperm quality decreased with age, but not in the 129ter. Oogenesis and fertility were affected in all dnd1- strains, with the mildest phenotype in 129dnd1- strain.

The results concerning the phenotypic differences between 129ter and 129dnd1- as well as the cross-breeding experiments with Nanos2- or Nanos3- are novel and shed light on the mechanisms of STT pathogenesis. Although the conclusion that ter mutation causes dndl1 loss confirms previous studies, this is the most rigorous confirmation of this hypothesis. The work is very comprehensive and technically very well done. The figures are very nice and informative. The findings are discussed carefully, without over-interpreting the data.

I congratulate the authors for their excellent work and think that the paper is ready for publication essentially as is.

Minor specific comments:

1. The studied species (mouse) should be mentioned in the title.

Response: Thank you for the suggestion. As per the reviewer’s suggestion, we have indicated the species studied by adding the word “Mouse” in the title.

2. The description of figures is included in the text of the Results and there are no separate figure legends. In my opinion this avoids redundant data repeating but the editors can decide whether or not this is OK.

Response: As per the journal’s formatting guidelines, we had placed the figure caption along with the corresponding legend directly after the paragraph in which the corresponding figure had been first cited.

Reviewer #2: Imai et al. introduce a new Dnd1Δ/Δ mouse line with a loss of function phenotype and show that double mutants of Dnd1 with Nanos2 and Nanos3 increase the testicular teratoma incidence. By comparing their knockout phenotype in different genetic backgrounds to the well-described Dnd1+/Ter mice, they show similarities in the phenotypes supporting the hypothesis that the Ter mutation leads to a loss of functional protein. By using different genetic backgrounds, they address the known influence of the genetic background on the DND1 mutations on the tumor incidence.

The authors present a well designed study. They showed appropriate controls and their conclusions are supported by statistical analyses. All methods are described in sufficient detail.

Minor points for improvement prior to publication:

- Figure presentation: The labeling of the graphs is relatively small and may be very hard to read when considering the usual figure size in the formatted manuscript.

Response: We have increased the font size of the labels used for the graphs in all the Figures.

- Line 105: What is the genetic background of the DND1flox/flox used to produce the Dnd1Δ/Δ mouse line? The cited publication (Suzuki et al. 2016) does not describe this specific line in detail.

Response: Thank you for highlighting this point. We had previously established a Dnd1_flox mouse line by using TT2 ES cells and maintained it via interbreeding to generate Dnd1flox/flox mice. We used these Dnd1flox/flox mice to produce the Dnd1_Δ mouse line. We have added this information in the “Mice” subsection of the Materials and methods section (Lines 104–111). 

- Line 526: In your discussion, you mention the observed differences to the study of Zechel et al. While I appreciate the given explanations, I was wondering how you ensured that the backcrosses still share the 100% of the genetics and that no other (de novo) mutations occurred explaining the striking differences between Dnd1Δ/Δ and Dnd1KO/KO mice?

Response: We understand the reviewer’s concern. To address this point in the manuscript, we have now mentioned the possibility that a de novo mutation could have occurred during the establishment of the Dnd1-KO mouse line, resulting in the observed striking differences (Lines 544–546).

- Line 540: You mention genes close to DND1, that may be regulated by deleted regions and potentially explain observed differences. Your point may be strengthened by giving specific examples, for example if neighboring genes play a known role in developmental processes affected or if the genetic organization (TADs, known enhancers) in this region are described in the literature and do confirm your hypothesis.

Response: To the best of our knowledge, no study has reported the physiological role of the neighboring genes in spermatogenesis. We have mentioned this in the Discussion (Lines 527 and 528). In addition, we have listed the genes neighboring Dnd1 and the phenotypes of their knockout mouse lines in Figure S5. 

Reviewer #3: The paper “Dead end1 acts with Nanos2 and Nanos3 to regulate testicular teratoma incidence” by Imai et al present functional analysis of the Dead end protein in mouse.

The authors present the phenotype of the “full knockout” of the gene and compare it with that observed in the Ter mutation where the protein is truncated.

The authors suggest only the development of the germline is affect in the mutant, which is different from the conclusion of a previous paper where Dead end was considered to be essential for embryonic development. The authors provide evidence that from the point of view of teratoma development the complete null and the Ter allele are similar. The authors also present the phenotype Dead end loss of function has on ovary development, which is a poorly studied topic. Last, the authors present a series of experiments should genetic interactions between Dead end and Nanos proteins, thereby contributing to the understanding of the molecular mechanisms of Dead end function.

Overall, the paper provides new data that is of importance for researchers in the field of reproductive biology and the work should definitely be published in the journal after addressing the following points.

1. The paper would benefit a lot from English editing. In many cases the message is understood only after reading certain sentences several times.

Response: Our manuscript had already been proofread, as indicated by the certificate provided on the last page of this file. However, in accordance with the reviewer’s comment, the manuscript was submitted for English language proofreading again after incorporating the changes suggested by the reviewers.

2. “In the BL6 strain, teratomas were not observed in the testes of all three genotypes, whereas teratomas developed in the testes of approximately 10% of the MCH Dnd1Δ/Δ male mice, indicating that the MCH strain has low sensitivity to testicular teratoma.” – reading this sentence one should would define the MCH strain as having HIGH sensitivity, since the comparison is to the BL6 strain that does not form teratomas. The authors should modify / correct this sentence.

Response: We understand the reviewer’s concern. We have revised this portion in the manuscript to avoid misinterpretation by the readers (Lines 233–239).

3. In Figure 1A- C the authors present examples for the testes phenotypes. The least severe difference between the wild-type and the mutant is observed in Figure 1C (for the 129 strain). From Figure 1D, the 129 strain shows the strongest effect. The authors should present in 1A-C examples that fit the average phenotype presented in Figure 1D.

Response: Thank you for your suggestion. However, we faced limitations in obtaining 129 Dnd1+/+, Dnd1+/Δ, and Dnd1Δ/Δ male mice in a litter by intercrossing 129 Dnd1+/Δ mice because 129 Dnd1+/Δ male mice show decreased fertility and eventually become sterile by 12 weeks of age, as indicated in Figure 3C. In addition, approximately 95% of Dnd1Δ/Δ male mice have testicular teratomas. Thus, we have a very low chance to obtain 129 Dnd1+/+, Dnd1+/Δ, and Dnd1Δ/Δ testes with no teratomas in a litter. From these reasons, we currently do not have another picture for Figure 1C and it would take much time to obtain new one. As pointed out by the reviewer, Figure 1C does not represent the average phenotype presented in Figure 1D; however, it fits in the data range. Therefore, we think that Figure 1C would not critically hinder the reader’s sound understanding of the information provided.

4. In Figure 1E-M, show magnified boxes in all cases not only in panels F, I and L. This way the phenotypes can be better appreciated.

Response: Thank you for your suggestion. We have now provided magnified insets for the remaining figure panels also in Figure 1E-M.

5. “Only the Dnd1Δ/Δ ovaries from the 129 female mice appeared to be slightly larger than those of the other two strains…”. If this data exists, provide quantitative information rather than “slightly”.

Response: We do not have quantitative data for this analysis because the Dnd1Δ/Δ ovaries were too tiny for accurate measurement of their weight. However, as the ovaries of the 129 Dnd1Δ/Δ mice were obviously larger than those of the other two strains (Figure 2A-C) and contained a considerable number of oocytes, unlike the ovaries of the other two strains (Figure 2L). We hope that the present data would not hamper readers’ understanding of the data being presented. 

6. In Figures 2D-L, provide magnification boxes and marked specific cell types / structures such that the information is more accessible for a broader readership.

Response: We have provided magnification insets for better visualization of the oocytes.

7. In Figure 3A, B, mark the mice as Nr 1 , Nr 2 and Nr 3 rather than No 1, No 2 and No 3. As it is, it looks like a gene name and not a mouse number. The best would be to use Mouse 1 , Mouse 2 and Mouse 3 as there is enough space for that.

Response: In accordance with the reviewer’s comment, we have changed the labels for the mouse number in Figure 3A–C and 4C.

8. The authors should try to reduce the text in the figure legends such that the information is not redundant with what is provided in the figure itself. For example in the legend of Figure 3- “The vertical axis represents the average number of offspring per copulatory plug, while the horizontal axis represents the age of the male mice.” The same information is provided in the Figure itself.

Response: We have deleted redundant information from the legends of Figures 2, 3, and 4.

9. “…were subjected to flow cytometric analysis performed using antibodies against PLZF, because… “ define PLZF in the text the first time it is mentioned and not only in the methods section.

Response: As per your instructions, we have now defined PLZF at the first instance it is used in the Results section (Lines 370 and 371).

10. The authors start a section with the sentence “We next checked whether the Dnd1-Δ mutant mice exhibited phenotypes other than those of Ter mutant mice.”. In this section the Ter mutant is not directly referred to. The authors should rephrase it.

Response: We have rephrased the sentence as “We next checked whether the Dnd1-Δ mutant mice exhibited phenotypes other than those mentioned above.”

11. “These results support the idea that Ter is a loss of Dnd1 mutation, which in turn suggests that the Dnd1-Δ mutant allele causes some defect leading to sperm count decrease and sperm malfunction independently of Dnd1 loss.” This statement is very strong and is not proven by the data provided. The assumption that Ter is a complete loss of function, which questions the specificity of the knockout phenotype relies on Western blots. As very low amounts of the shorter version of Dead end are not necessarily detected, it could in principle be that the milder phenotype of Ter results from low amounts of the truncated version of the protein. In addition, it could be that the level of the Ter protein is higher at stages that differ from those examined in the Western blots. This point should be addressed and the statement the authors put forward should either be deleted, or presented as one of several options.

Response: To address the point raised by the reviewer, we have revised the sentence “These results support the idea that…” to “These results support the possibility that…”. In addition, we have mentioned the possibility that DND1Ter is still expressed and that this protein alleviates the phenotype of Dnd1 loss causing the phenotypic difference between Δ and Ter in the Discussion section (Lines 505–514).

12. “In addition, introducing the heterozygous Dnd1-Δ mutant allele into both Nanos2+/LacZ and Nanos2LacZ/LacZ male mice increased the ratio to 38.8% and 80.0% (Fig. S4A)..” – the information is not. Provided in Figure S4A. Perhaps the authors mean Figure 5a here.

Response: We are afraid that our figure citations might have led to some confusion. To avoid this confusion, we have now cited Fig. S4A along with Fig. 5A (Line 456). Similarly, we have now cited Fig. S4B along with Fig. 5B (Line 464).

13. In Figure 5 the lack of germ cells in homozygous nanos3 mutants is determined by expression of daz and nanog. The authors conclude that the cells underwent apoptosis, but do not show it directly. The loss of cells could for example be transfating etc. The sentence should just be rephrased or the apoptosis claim deleted.

Response: As per the reviewer’s comment, we have rephrased the sentence mentioning the possibility of transdifferentiation to somatic cells (Lines 488–492).

---

## [Decision Letter · Decision Letter 1]

7 Apr 2020

Mouse Dead end1 acts with Nanos2 and Nanos3 to regulate testicular teratoma incidence

PONE-D-20-03327R1

Dear Dr. Suzuki,

We are pleased to inform you that your manuscript has been judged scientifically suitable for publication and will be formally accepted for publication once it complies with all outstanding technical requirements.

With kind regards,

Stefan Schlatt

Academic Editor

PLOS ONE

Additional Editor Comments (optional):

Reviewers' comments:

Reviewer's Responses to Questions

**Comments to the Author**

1. If the authors have adequately addressed your comments raised in a previous round of review and you feel that this manuscript is now acceptable for publication, you may indicate that here to bypass the “Comments to the Author” section, enter your conflict of interest statement in the “Confidential to Editor” section, and submit your "Accept" recommendation.

Reviewer #1: All comments have been addressed

Reviewer #2: All comments have been addressed

2. Is the manuscript technically sound, and do the data support the conclusions?

Reviewer #1: Yes

Reviewer #2: Yes

3. Has the statistical analysis been performed appropriately and rigorously? 

Reviewer #1: Yes

Reviewer #2: Yes

4. Have the authors made all data underlying the findings in their manuscript fully available?

Reviewer #1: Yes

Reviewer #2: Yes

5. Is the manuscript presented in an intelligible fashion and written in standard English?

Reviewer #1: Yes

Reviewer #2: Yes

6. Review Comments to the Author

Reviewer #1: Thank you for this excellent and well written paper. I have no additional comments.

The paper is ready for publication.

Reviewer #2: Dear authors.

Thank you for submitting the revised manuscript. My questions were answered approprietly and the suggested corrections were included in the manuscript.

7. PLOS authors have the option to publish the peer review history of their article (what does this mean?). If published, this will include your full peer review and any attached files.

Reviewer #1: No

Reviewer #2: No

---

## [Editor Report · Acceptance letter]

10 Apr 2020

PONE-D-20-03327R1 

Mouse Dead end1 acts with *Nanos2* and *Nanos3* to regulate testicular teratoma incidence 

Dear Dr. Suzuki:

I am pleased to inform you that your manuscript has been deemed suitable for publication in PLOS ONE. Congratulations! Your manuscript is now with our production department. 

With kind regards,

on behalf of

Dr. Stefan Schlatt 

Academic Editor

PLOS ONE